

**Catastrophic debris flow triggered by an extreme rainfall event in the Volcán village,**
**January 2017. Cordillera Oriental of Argentina**
María Yanina Esper Angillieri[1], Laura Perucca[1,2] and Nicolás Vargas[2]
[1]CONICET- CIGEOBIO. Facultad de Ciencias Exactas, Físicas y Naturales, Universidad
Nacional de San Juan. yaninaesper@gmail.com
[2] Gabinete de Neotectónica y Geomorfología, INGEO. Facultad de Ciencias Exactas, Físicas y
Naturales, Universidad Nacional de San Juan. Departamento Geología. Facultad de Ciencias
Exactas, Físicas y Naturales, Universidad Nacional de San Juan. lperucca@unsj-cuim.edu.ar,
nicolasvargas2003@yahoo.com.ar
**ABSTRACT**
Slides, rockfalls, debris floods and debris flows are periodical events in the dry mountainous
regions of Argentina, during times of torrential rainfalls. In the Grande River basin, Jujuy
Province, these processes take place almost every summer. Extreme rainfall on January10, 2017
caused the seasonal acceleration of large-scale and slow-moving landslides in the Los Filtros
River basin. These slides broke down into a disaggregated mass, triggering a debris flow which
transformed progressively downstream into a debris flood, producing widespread damage along
a narrow valley (named Quebrada de Humahuaca), with the Volcán village withstanding the
worst of the disaster. The event caused four fatalities and great economic losses, mainly
destroying infrastructure and buildings. In order to document this catastrophic event and to
explore its causes, a morphometric analysis of the Los Filtros river basin, tributary of the
western margin of the Grande River and located on the Cordillera Oriental area, was carried
out. The drainage network was derived from digital elevation models. In addition, some
landslides were mapped using high-resolution satellite data acquired before and after the event.
Of a total landslide area of 2.39 km$^2$, 0.60 km$^2$ was considered as active and 0.089 km$^2$ as new



sliding area (from 2015 to 2017) associated to the large-scale and slow-moving landslides. The
geological characteristics of the study basin are very favourable conditioning factors in
landslide generation. Precambrian- age low grade metaclastics shatter in the frost climate of the
higher mountains and poorly consolidated Quaternary deposits along the sides of the gully
erode readily and become source material for landslide that damage or bury roads, railroads,
and houses. Finally, this study aims to increase knowledge of all the above-mentioned events
in order to provide several methods of analysis for landslide prevention and control.
**Keywords:** Debris floods and debris flows, basin morphometry, Slow-moving landslides,
Jujuy, Argentina
**1. Introduction**
Landslides sometimes occur on mountain slopes triggered by heavy rains changing into debris
flows and then moving into mountain rivers, in a complex process. As was pointed by Hungr
et al. (2013) shallow slides may begin with slow pre-failure deformation and cracking of
surficial soil on a steep hillside. Then, landslide mass accelerates, disintegrates, enlarges
through entrainment and becomes a flow like debris avalanche that enters a drainage channel,
entrains water and more saturated soil and turns into a debris flow. When slope diminishes, the
flow drops the coarsest fractions continuing as a sediment-laden flood. These authors proposed
to apply the simple traditional term "debris flow" to the whole scenario. Additionally, these
floods usually occur in mountain river basins draining less than 1000 km$^2$ (Gaume and Borga,
2008; Lumbroso and Gaume, 2012).
Debris floods and debris flows that take place in almost all the tributary valleys of the Grande
River are the main hazardous processes that affect this portion of the Cordillera Oriental
foothills of the Jujuy province in northwestern Argentina, inducing serious consequences on
the erosion and sedimentation activity along the Quebrada de Humahuaca (Cencetti et al., 2001)



where numerous human settlements are located (Fig. 1a, b). This is due to the special
morphometric, geographical and geological configuration of river basins in the Cordillera
Oriental of Argentina that is extremely favourable for the generation of debris floods and flows.
Moreover, a great amount of loose debris, consequence of slope processes such as slides and
rock falls, is available due to the geological characteristics of the outcropping lithology and
structures. Debris flows/floods pose a serious threat to the socio-economic and physical
environment of this region. In Table 1 are listed some of the most catastrophic events
responsible for most deaths and damages to roads and villages that have occurred in
northwestern Andes of Argentina.
Morphometric characteristics of a river basin area unit are basic tools to estimate and predict
its behaviour under conditions of heavy rainfalls, and to compute the potential hazard of debris
flows/floods to downstream settlements and infrastructure. For this reason, morphometric
analyses were used for river basin characterization from different areas of the world in several
previous researches, such as Topaloglu (2002), Moussa (2003), Sreedevi et al. (2004, 2013),
Srinivasa Vittala et al. (2004), Mesa (2006), Esper Angillieri (2007, 2008, 2012), Esper
Angillieri and Perucca (2014a,b) and Perucca and Esper Angillieri (2011), among others.
The focus of this work is to describe and analyse the destructive event that occurred in the
Volcán village on January 2017, studying some geomorphological and hydrological aspects and
identifying their effects during torrential rains in order to generate latest information of the
basin for future river basin management.
**2. Study area**
The Volcán village, with 1731 inhabitants (2010), is located 41.9 km north of San Salvador de
Jujuy city (capital town of the Jujuy province), Argentina. In this area operates since the 1970s,
a cement production plant that is the main supplier of cement of the region and southern Bolivia.
(Fig. 1a-c)





The Los Filtros River, which crosses the Volcán village, is a tributary of the northern margin
of the Grande River that flows to the south in a narrow mountain valley trending N-S (Fig. 1d).
This valley, named Quebrada de Humahuaca, follows the line of a strategic route connecting
Argentina to Chile and Bolivia (National Route 9). There are almost 17 villages along the
Grande River valley, which is characterized by a variable discharge due to extreme climatic
variability, both in space and time. Mean daily discharge is between 16.4 and 24.75 $m^3$/s, with
a maximum of 74.56 m³/s in February and a minimum of 4.83 m³/s in October. The historical
maximum discharge was 358$m^3$/s and the minimum was 3$m^3$/s (Paoli et al., 2011).
The annual average temperature in the region is approximately 14°C; July is the coldest month,
with an average temperature of 5.2°C, and the hottest month is December, with temperatures
averaging 19°C (Buitrago, 1999).
Tropical humid air masses of Atlantic origin transported by the South American Summer
Monsoon influence regional climate of NW Argentina. It is characterized by the large
seasonality with most of the total annual precipitation falling in austral summer, from December
to April (Bianchi and Yañez, 1992; Garreaud and Aceituno, 2007), with an average yearly
rainfall of more than 400 mm. The total annual rainfall registered in the Volcán locality, from
1934 to 1996, oscillates from 123 to 719 mm, with an average of 391.31 mm, while the
maximum monthly average rainfall is recorded in January with values close to 115 mm (Fig.
1e). The largest amount of rainfall measured in one month was 271 mm, in February 1971 (Data
from INTA EEA SALTA © Copyright 2002).
On January 10, 2017, between 8 a.m. and 10 a.m. (local time), a very torrential rainfall of 170
mm (according to Engineer Sadir - Water Resources Director, verbal communication) affected
the Los Filtros River basin.
Numerous slides were generated on the slopes of the headwaters of this drainage basin,
initiating a flow where the main stream discharged quickly with a high sediment transport. At



9:20 a.m. (local time), the debris flow/flood descended downstream from west to east, breaking
the defenses and crossing the National route to the Volcán village. Citizens considered this
event as the most catastrophic one to occur in 40 years, causing 4 fatalities, more than a
thousand evacuees and great economic losses along the National Route N° 9 (Figs. 2 a-d and
3a-c). As the locals explained, the most affected area was the closest to the San Martin Street,
located in the north of the village and with a W-E orientation, where the mud reached the
houses' roofs, burying trees and light poles (Figs. 2c, d and 3c, d). Blocks and mud rushed
through the village burying roads, vehicles and houses and destroying or damaging most of the
local streets and shops (Figs. 3d-f).
The material deposited with a variable width of about 300 to 500 m and an average thickness
of 1.5 m, became very fine in the front of the flow, acquiring a viscous behaviour and decreasing
its speed. The deposits height reached up to 2 m according to the splash marks and material
deposited. The muddy nature of the mass made it difficult to remove the material during the
cleaning of the streets. In addition, the debris flow/flood reached the channel of the Grande
River, partially obstructing it (Figs. 3g, h).
**2.2 Geological setting**
The Los Filtros River is located in the Cordillera Oriental geological province, consisting of
large folds (Mon and Salfity, 1995), with Precambrian-age rocks of low metamorphic grade
cropping out in the cores of the anticlines, beneath folded sedimentary strata of Cambrian,
Ordovician, and Cretaceous age. Over these units, there is an extensive Quaternary cover
generated by gravity processes acting in the upper part of the slopes together with debris flows
and alluvial deposits in the valley bottom (Chayle and Aguero, 1987). Precambrian phyllites
constitutes the bedrock geology of the upper portion of the Los Filtros River basin. These rocks
overlay the cretaceous sandstones by a reverse fault that strikes NNW and dips to the west, with


probable Quaternary tectonic activity. This regional fault system has an east-vergence, trending
N-S to NNE. In the lower basin, mainly Quaternary alluvial deposits are exposed (Fig. 4a-e).
The vegetation cover in the basin is characterized by shrub steppe vegetation that is plentiful as
well as cacti, mainly cardones (*Echinopsis atacamensis*), dwarf forests and bromeliad cushions.
Above 3400 m asl, dominant vegetation consists in shrub steppe, grassland steppe, queñoa
(*Polylepis tarapacana*) forests and in the wet areas and spring sources, pastures called vegas
(Frigoni Prado, 2014).
Therefore, the soil can undergo intense superficial erosion during high intensity rainfall events.
**3. Materials and methods**
The analysis of the event was carried out through the compilation of local newspaper reports
and field investigation. Furthermore, Los Filtros river basin delineation and the morphometric
characterization through topographical data and satellite imagery were made. Post-disaster Spot
images provided by CONAE (Comisión Nacional de Actividades Espaciales, Argentina) ©
CNES 2017, Distribution Spot Image SA were compared with pre-disaster images, in order to
explore the overall scenario of the event. Previous small slides related to large-scale and slow-
moving landslides were detected by using a semi-automatic analysis of the variations in the
spectral signature of the land surface and resample with ESRI's ArcGis 10.3. Large-scale and
slow-moving landslides were identified using high-resolution satellite imagery from Google
Earth™, which was georeferenced to a geographical coordinate system (WGS84) within a
geographical information system (GIS). The basin was delineated based on the water divide
line concept and was on-screen digitalized using the same GIS technology. The main channel
length (Mcl) and length (L) were calculated according to Schumm (1956).
The Elevations, Slope, Topographic wetness index and the Sediment transport capacity Index
were obtained using a digital elevation model provided by ALOS/PALSAR





ALPSRP268846700 (ASF DAAC, 2015) with 12.5 x 12.5 m spatial resolution. The slope
analysis was performed in a GIS environment.
The morphometric parameters of the basin, which divided in basic parameters are area (A),
perimeter (P), length (L), mean width (W), maximum and minimum heights (H, h) and main
channel length (Mcl), were quantitative calculated using GIS. Besides, several derived and
shape morphometric parameters were obtained using the equations in Table 2, like circularity
index, elongation ratio, form factor, sinuosity index ratio, relief ratio and basin relief, among
others. These relief properties in the morphometric analysis bring into consideration the
influence of aspect and height over the river basin area.
The geologic map modified from Savi et al. (2016) was used to construct a representative
longitudinal topographic profile with the distribution of the main knickpoints along the Los
Filtros River with SAGA GIS, in order to show regional topographic features controlling the
river.
**4.  Result and discussion**
**4.1. Catastrophic event of January 2017 and the Los Filtros River basin description**
The Los Filtros River flows along the eastern flank of the Chañi Hill (elev. 4,139 m asl), passing
the Volcán village (elev. 2,125 m asl), to its junction with the Grande River (elev. 2,075 m asl).
The results of the morphometric analysis of this mountain river basin is given in
Table 2, where the circularity index, elongation ratio and form factor show a very elongated
basin. Basin morphology (Table 2) may be used to differentiate between basins prone to floods,
debris floods and debris flows (Jackson et al., 1987; Wilford et al., 2004). Thus, the debris flow
catchments have Melton's ruggedness number >0.6. Basin constituted by fine-grained
materials, such as the Los Filtros River, may, however, have a lower Melton Ratio and still be



prone to debris flow, since these materials are easily mobilized and can then travel longer
distances.
Small (<20 km$^2$), rugged and low-order basins produced small and steep fans dominated by
debris flow processes implying different sediment-water mixtures (Pierson, 2005). Such is the
case of the study area, a system of distribution of rainwater in a rather small reception river
basin, with a main discharge channel that is excavated in a very narrow valley with almost
vertical walls of up to 50 m of height. The Los Filtros upper river basin is located almost 4,000
m high, in a hyper arid environment that is only disrupted by very heavy rainfall during summer.
Throughout the year, frequent mass removal processes take place, such as slides and large
blocks falls from loose, fractured and weathered materials. When heavy rains occur, they can
re-mobilize large amounts of debris carried by high-density flows in a main river collector.
According to the movement mechanism and the genesis and plasticity of the material, the flow
that occurred in the upper sections of the Los Filtros River basin can be classified as a non-
plastic debris flow, with the material deposited in a steep channel (Hungr et al., 2001). The flow
mobilized a mixture of mud and medium blocks (mostly between 10 and 20 cm in diameter),
with few blocks > 1 m in diameter. The rocky substratum of the area composed of sandstones,
limestones, metamorphic rocks and colluvial deposits of the river basin, saturated by the intense
precipitation, began to move downstream.
We made a brief description of debris deposits with thicknesses ranging from 0.5 to 1.5 m in
three natural and artificial exposures located along the Los Filtros River, two weeks after the
event. One outcrop is located upstream, near the national route 9, the second some meters
downstream of the route and the third almost at the mouth of the debris flow, near the
confluence with the Grande River. Debris flow deposits vary from west to east. Upstream, the
deposit mainly consists of rocks with a grain size of 10–40 cm. The cobbles and boulders are
angular-shaped and unsorted and their size decreases significantly downstream while the clay

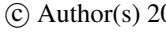


content increases. Blocks are clast-supported, with a very low content of a sandy-clayed matrix
that increases the floatage of the clasts and lubricates them to enable their transport.
Downstream, the thicknesses (up to 1.5 m) of a poorly sorted mass of coarse material was
measured, resulting in approximately 70 % gravel and cobbles, 20 % sand and 10 % lime-clay.
According to the plasticity of this material, the flow was classified as a debris flow. Further
downstream, the fabric deposit is disorganized, with the larger clasts dispersed within a
predominantly clayey matrix (matrix-supported) (Fig. 3f). These materials might have been
deposited by an intermediate type of flow that corresponds to those with low viscosity and high
density (Pierson, 2005). They belong, at least in part, to the so-called debris flood, where the
slope decreases markedly, especially near the confluence of the Los Filtros River with the
Grande River (Figures 3g, h). Deposits in this sector possess mainly pebbles and a greater
percentage of an argillaceous matrix (~40%) that resemble lava flows, with longitudinal furrows
and ridges in its frontal lobe (Figs. 3g, h).
Harrington (1946) reported that Volcán area debris/ mud flows advanced rhythmically making
jumps of 5-10 m every few seconds at velocities of 10-15 km/h. The larger fragments were
deposited near the apex of the fan and the finer materials in the border more near the horizontal
outer area of the fan. On the other hand, Polanski (1966) also described in the El Volcán area
rhythmically advancing debris flow events in the form of internal waves with a ~ 15 km/h speed
over 15 km, with some rapid and short-lived fluid episodes, locally showing a considerable
thickness of sediments. According to several witnesses of the event, the debris flow that
occurred in January 2017 had the same characteristics to those observed by these authors, i.e
the larger clasts upstream national route 9 and a muddy, fluid and rhythmic behavior
downstream, clearly manifested in the preserved buildings covered by up to 2 m of mud and
debris carried by the flow, but without signals of destruction (Fig. 3d).





Gradient variations can be seen along the longitudinal river profile, with a concave shape
upstream, slightly concave in the middle channel, and a relatively straight profile (very low
concavity) at the end of the stream (Fig. 4f). Local distortions in the longitudinal profile
represented by knickpoints are mainly due to lithological contrasts between the Precambrian
metamorphic rocks, the Mesozoic sedimentary strata and the unconsolidated Quaternary
alluvial deposits. However, a structural control is not ruled out.
The elevation map of the Los Filtros River shows the distribution of altitudes in meters along
the basin, revealing a steep gradient oriented W-E (Fig. 5a).
The sediment transport capacity index SL (the distance from where the flow is originated, along
its path, to where it concentrates or deposits) of the basin ranges from 0 to 376.8 (Fig. 5b). The
larger the SL, the more water accumulates at the bottom of the field, increasing erosion.
The Topographic wetness index (TWI) was used in order to describe the effect of topography
on the location and size of saturated areas of runoff generation (Nefeslioglu et al., 2008; Akgun
and Turk, 2010). The TWI values calculated for the basin vary between 2.65 and 19.23 with a
mean value of 5.89 (Fig. 5c). This indicates the probable existence of saturated soil conditions
during rain events and the sediment accumulation (Beven and Kirkby, 1979). A comparison
between the obtained TWI (Fig. 5c) values and the landslide occurrence showed a coincidence
in the saturation and/or accumulation of material areas (Fig. 5d). This may be the result of the
availability of lithological units with a relatively high permeability and low surface runoff.  This
map shows similar results to the sediment transport capacity index SL.
The slopes map shows gentler slopes in the headwaters of the basin and a bedrock with typically
high slope angles and steep morphology, mainly in the hillside, with maximum slopes of 60°
and an average of 29° (Fig. 5e). This gives a good indication of the areas that correspond to
bedrock. The low slopes area in the end of the Los Filtros basin is the consequence of a repeated
sequence of debris flows.





Landsat 8 OLI images provided by CONAE taken on 3 January 2017 were employed, at a
spatial resolution of 30 m, to extract Normalized Difference Vegetation Index (NDVI) (Deering
et al., 1975). The vegetation density was determined by NDVI in order to observe its
relationship with a landslide area. The obtained values were on the closed interval [−1, +1].
Values approaching to +1 indicate dense vegetation while the values computed close to –1
indicate the lack of vegetation or bare lands. In the basin area, NDVI values were computed
between 0.0611 and 0.8061 (Fig. 5f). Considering their spatial distributions, we concluded that
approximately 76% of the study area is covered by dense vegetation with values of NDVI
ranging from 0.3 to 0.8, and a small portion of approximately 6% is covered by rock, coincident
with the sliding basin area. Nevertheless, due to the seasonal flora and the lack of perennial
species, the vegetation cover as contributing factor is relative, depending on the season (Fig.
5f).
**4.2. Sediment Sources and Supply**
In some small low-order basins, located in mountain environments, a dramatic response to large
flows is expected. In high relief areas such as first- and second-order basins, debris slides are
important geomorphic processes that can drastically change the drainage network system.
These kinds of events can occur in all climatic regions and should be considered as potentially
devastating natural hazards (Honer, 2010).
Phyllites and unconsolidated alluvial deposits are relevant to debris flow process in the basin,
as these rocks range in competency from slate or phyllite to metamorphosed pebble
conglomerates that degrade to fine sands. The resulting product constitutes a significant source
of relatively fine-textured sediment, easily mobilized in the channel, and capable of long run
out distances due to its texture. That is why the current geomorphic activity contributing to the
torrent's recharge with debris, is concentrated exclusively in the headwater.



In addition, the river basin is strongly affected by large-scale and slow moving landslides during
intense rainfall periods. Several landslides were confirmed to be present before the 2017 event,
based on pre-disaster images (Figure 6a-c). Nearly all of them showed signs of enlargement or
remobilization during the catastrophic event according to the post-disaster images (Fig. 6a'-c').
From the 2.39 km$^2$ identified as landslides area, 0.60 km$^2$ were classified as active and 0.089
km$^2$ as new slope failure area (from 2015 to 2017), associated to large-scale and slow-moving
landslides. In some sectors, the crown of an inactive landslide retreated 53 m and the width near
the crown increased from 58 to 89 m (Fig. 6c-c'). Comparison between satellite images taken
before and after January 10, 2017's event let the authors find its initiation site and the main
sediment-supplying zone in the upstream basin.  Besides, figure 6d-d'compares pre-disaster
scenario in the downstream river basin, with post-disaster situation after the January 2017,
resulting in the widespread destruction of the Volcán village. Material from a large-scale
landslide located in the middle of the basin, with several small-scale slope failures near the toe
(mainly coarse rock fall debris, and small debris slides) constituted an abundant supply of loose
debris that, lubricated by the rainfall and/or because of the erosion at the base were removed
and incorporated into the riverbed that result in a debris flow (Fig. 7a, b). We selected this large-
scale landslide in order to exemplify the major source and sediment supply to the river basin
(Fig. 7c). The landslide covers an area of 0.49 km$^2$ with slope height of 419 m (from 2352 to
2771 m asl), a width of 802 m and a length of 984 m. At the toe of the slope a large bulge
suggests that the river was dammed in the past (Fig. 7c).
**5. Conclusions**
Heavy rains occurred on January 10, 2017 caused the acceleration of large-scale and slow-
moving landslides that triggered a debris flow/flood in the Los Filtros river basin traversing the
National 9 trunk road and damaging the Volcán village. This event was favoured by the



topographic and geological characteristics (gradient, lithology, sediment capacity, lack of dense
vegetation, etc.) of the river basin that conditioned landslide generation that triggered a debris
flow/flood. The slopes of the Los Filtros river basin show large-scale and slow moving
landslides during torrential rainfalls providing loose debris to the riverbed that result in debris
flows.
Most of the pre-existing alluvial fans are settlement areas and for several generations, people
have lived in these landforms. Almost every summer, the slope failures interrupt the National
Route 9, one of the main routes between Argentina and Bolivia. As a result of the impossibility
to predict such events the evacuation of the population may result difficult. One way to mitigate
the effects of debris flood and debris flows would be to allow only crops in high-risk areas, to
reduce the harm to the population in case of a destructive event. Thus, it is necessary to make
detailed hazard zonation maps with an inventory of landslides, size, activity, among other
aspects. These studies are essential for an adequate land-use planning in mountainous areas.
Finally, it is necessary to increase the existing knowledge of such events to provide specific
skills and technical solutions for floods and debris flows prevention and control.
**6. Acknowledgements**
The authors acknowledge reviewers and funding received from PID 0799 of Consejo Nacional
de Investigaciones Científicas y Técnicas (CONICET) to support this research and CIGEOBIO
by providing funds for ArcGIS 10.3 software license. For this study, the authors requested
SPOT and Landsat satellite data to the National Commission on Space Activities - CONAE.



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

**Figure captions**
Figure 1. a) Location of the Jujuy province in Argentina; b) Main villages of the Quebrada de
Humahuaca; c) Map of Jujuy and Salta Provinces showing, in red dots, main affected areas,
detailed in Table 1; d) Los Filtros river basin, Volcán Village and National route 9 locations.
Surrounded by a black outline, the 2017 debris flow affected area; e) Average monthly rainfall
for the Volcán village area, data from INTA EEA SALTA©Copyright 2002.





Figure 2. Aerial oblique views; a) To the south, showing the cut of the National Route 9; b) To
the south, showing damages on the route and in the northern sector of the town; c) To the north,
showing damages in the cement factory, route and town and d) To the northwest, showing the
affected area in the town and the debris flow entering into the Grande River channel.
Figure 3. a) Westward view of the Los Filtros River basin. In the foreground, we can see the
deposits carried by the flow; b) View to the east, downstream of the Los Filtros River basin,
towards the town of Volcán; c) Damage to houses located west of National Route 9, d) Damages
in the houses located along the San Martín street. It is possible to appreciate the mark that
reached the flow in the walls of the buildings, e) Housing devastated by the flow; f) Detail of
the debris deposit. The average size of the blocks is approximately 20 cm; g) View to the south
of the debris flood deposit partially obliterating the riverbed of the Grande River; h) Detail of
the previous photograph. Yellow arrows indicate furrows and ridges formed in the front of the
flow because of the greater fluidity there.
Figure 4. a) Geological map of the Los Filtros River basin area (modified from Savi et al.,
2016); b) Precambrian phyllites outcrops; c) Paleozoic limestones; d) Mesozoic sandstones; e)
Quaternary alluvial deposits and f) Longitudinal the Los Filtros River profile.
Figure 5. a) Digital Elevation Model; b) Sediment transport capacity index (SL); c) Topographic
wetness index (TWI); d) NDVI; e) Slope map; f) TWI and post-disaster Spot image of the Los
Filtros River basin (Includes information © CNES 2017, Distribution Spot Image S.A., France,
all rights reserved).
Figure 6. Pre-disaster and post-disaster Spot image of the Los Filtros river basin showing
significant changes of the drainage after January 10, 2017 (Includes information © CNES 2017,
Distribution Spot Image S.A., France, all rights reserved). a) and a') northern high basin; b) and
b') south high basin; c) and c') middle basin, in circle is shown, as an example, a reactivated



small slide, associated to a large-scale landslide, where the crown has increased from 58 to 89
m; d) Volcán village and National route pre-disaster and d') post-disaster.
Figure 7. a) Pre-disaster and b) Post-disaster Google Earth image of the Los Filtros River
medium basin showing the selected large-scale a slow-moving landslide; c) Schematic model
of main sediment sources and supply in the basin associated to large-scale landslide (concept
after Hasegawa et al., 2008).
Table 1. Most catastrophic events occurred in northwestern Andes of Argentina (mainly Jujuy
and Salta provinces) in the last century
Table 2. Morphometric parameters of the Los Filtros River basin.

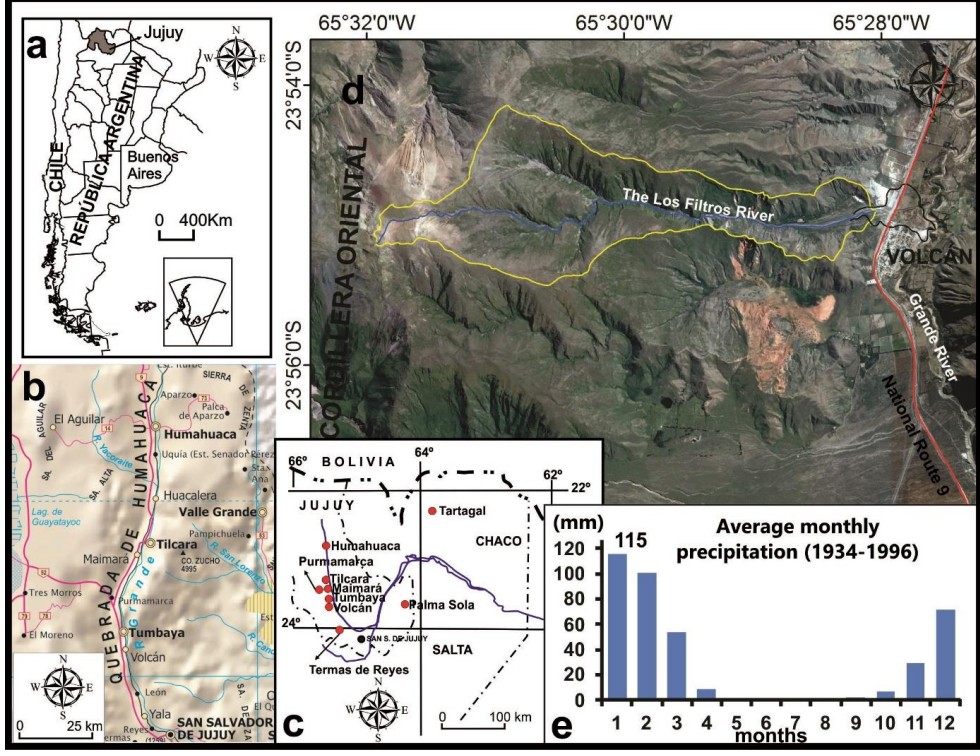



1    Figure 1

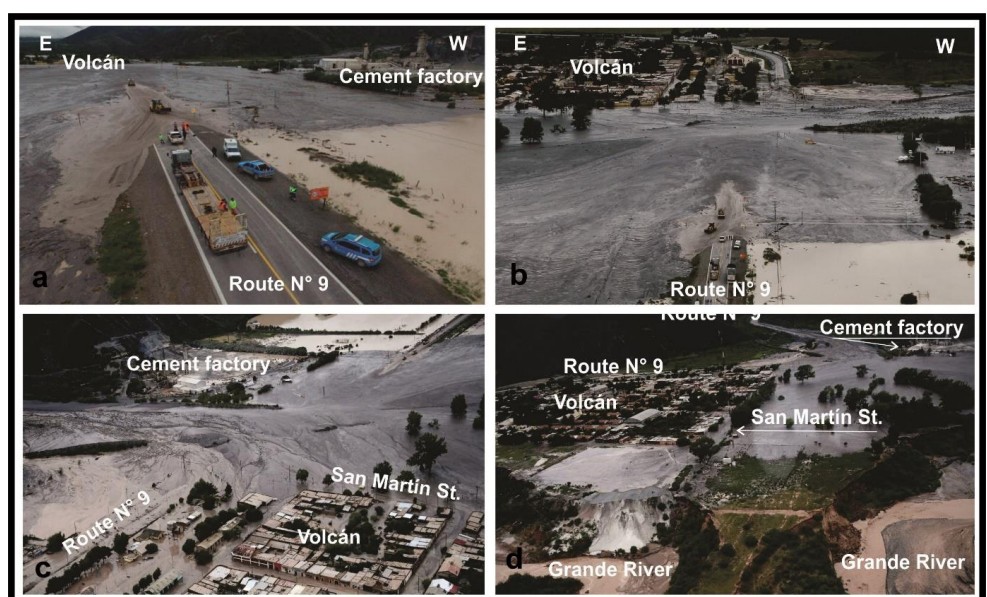

3    Figure 2





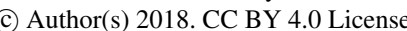

Figure 3

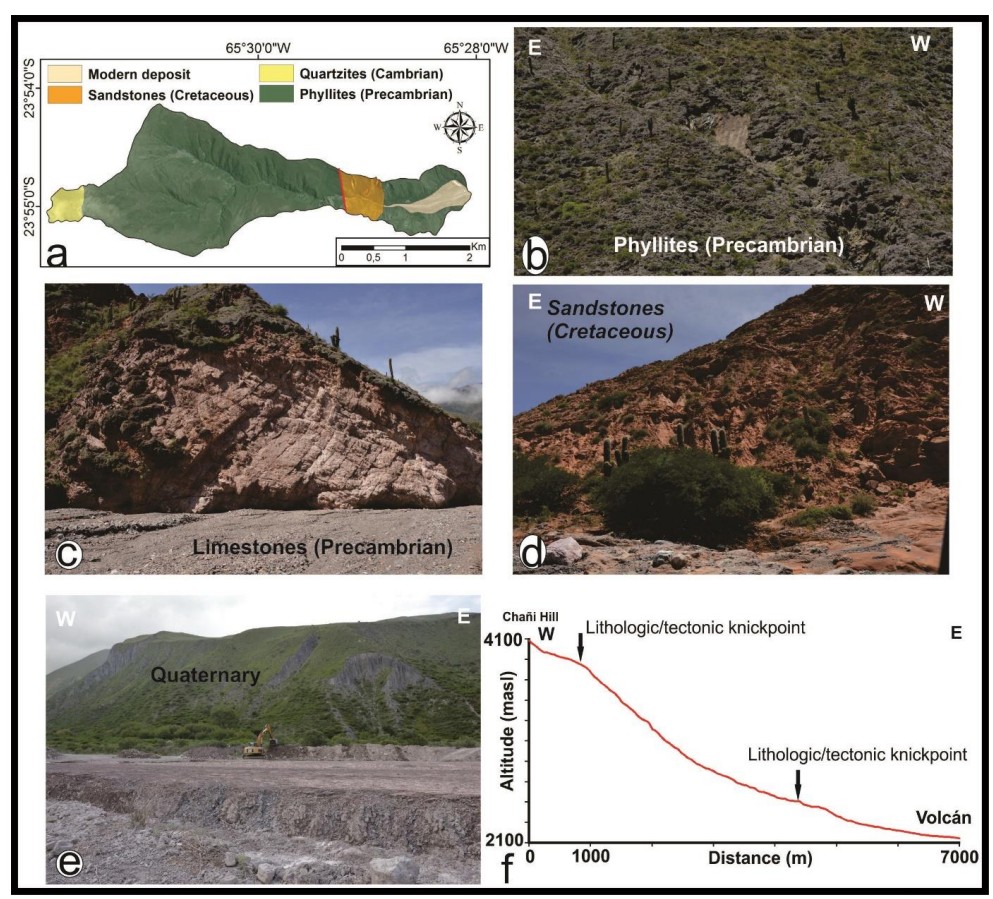

2     Figure 4



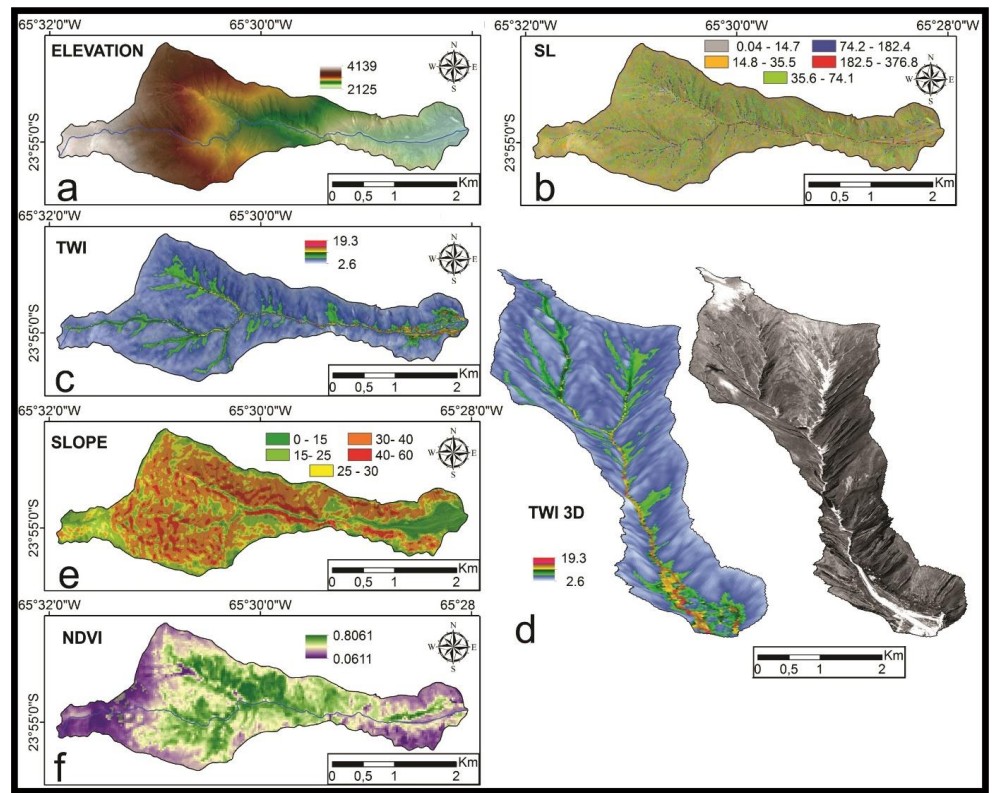

2    Figure 5



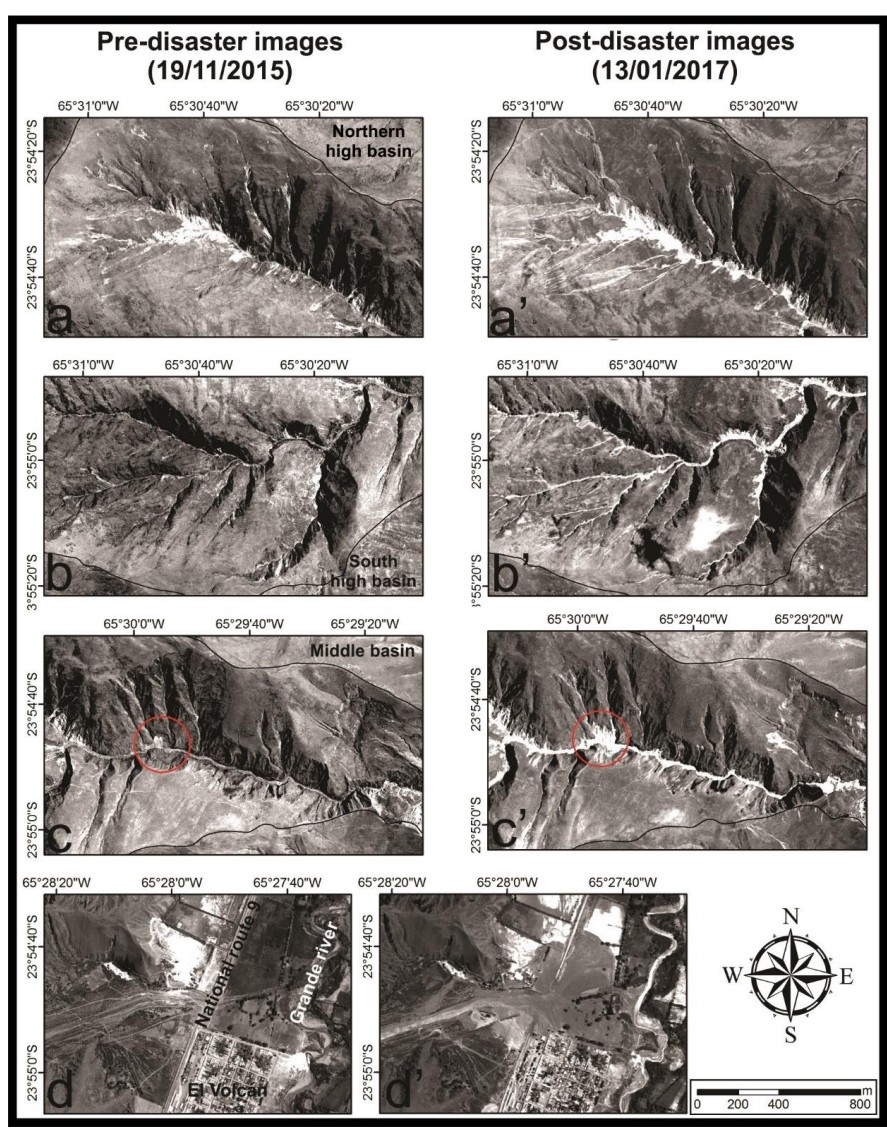

2    Figure 6

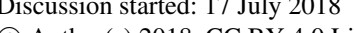



2  Figure 7


| Date | Region affected | Fatalities and damaged caused | Mechanisms involved | References |
|---|---|---|---|---|
| March 2, 1945 | Quebrada de Humahuaca - Volcán village | The Grande River dammed forming a 2 km long lake that flooded the Volcán village. Town destruction. Loss of agricultural production. Isolation and food shortage. | Debris flood | Castro (2001) |
| February 18, 1969 | Tumbaya village, Jujuy province | The event interrupted the road and rail communication. Damages to the cemetery and town. | Debris and mudflows | Castro (2013) |
| January 11, 1974 | Tilcara, Humahuaca, Tumbaya and Maimará villages, Jujuy province | 82 evacuated, 167 affected, 26 ha destroyed (most in Tilcara); Losses for farmers. Declaration of state of emergency throughout Provincial territory. | Debris flood | Castro (2013) |
| Summer, 1983 | Tilcara village, Jujuy province | Two tourists lost their lives and the deposits dammed the Grande River valley, forming a lake. | Debris flood | Cencetti et al. (2004) Marcato et al. (2009) |
| February 2, 1984 | Purmamarca, Tilcara; Tumbaya and Humahuaca villages, Jujuy province | Interruption of communications (the rail service remais interrupted for two months). 13 fatalities. Purmamarca station buried by the mud. Destruction of homes, 150 evacuated. Loss of crops, 100Ha were affected by the destruction of irrigation channels. | Debris and mudflows | Castro (2013) |
| February 2, 1985 | Volcán, Purmamarca, Maimará, Humahuaca villages, Jujuy province | More than 4 m of thick deposits covered a large area, resulting in the complete destruction of a railway bridge, the interruption of the National Route 9. Railroad tracks destroyed. Partial flooding of Humahuaca localities (more than 200 houses affected, more than 50 evacuated). In Maimará there is an 80% loss of crops. | Debris flow | Cencetti et al. (2004) Marcato et al. (2009) Castro (2013) |
| April 4, 2001 | Palma Sola town, Jujuy province | Severe damage to the irrigation infrastructure, land, people and buildings. | Debris flood | González Díaz and González (2002) |
| April 4, 2006 | Tartagal city, Salta Province | The event destroyed or severely damaged bridges and roads along the course of the Tartagal River, which caused the city of Tartagal to be practically isolated. | Debris flood | Latrubesse and Brea (2009) |
| March 7, 2007 | Purmamarca, Jujuy province | The design capacity of bridges was drastically reduced by the obstruction caused by the uprooted trees. | Debris flow | González et al. (2009) |
| March 25, 2007 | Purmamarca, Jujuy province | The event devastated everything existing along its way to Purmamarca. | Debris flow | González et al. (2009) |
| February 9, 2009 | Tartagal city, Salta province | The event dragged a large, unusual amount of sediment and trees. The collapse of a railway bridge caused the blockage of the riverbed, flooding the city. 3 dead, 600 evacuated and 10,000 injured | Debris flood | Brea et al. (2013) |
| January 12, 2010 | Comedero River, Termas de Reyes, Jujuy province | Devastated everything existing along its way, causing 87 injured and severe material damage. | Mud flow | González et al. (2012) |
| December 11, 2012 | Volcán village, Jujuy province | Affected the Volcán village, damaging the National Route 9 and several buildings. | Debris flow/flood | El Tribuno Newspaper (2012) |
| January 10, 2017 | Volcán village, Jujuy province | The event raced down the Los Filtros river basin until it finally arrived at the Volcán village leading to great destruction. 4 deaths | Debris flow/flood | The present research |

Table 1


*The Los Filtros river basin*

| A [km²] | P [km] | L [m] | H [m asl] | h [m asl] |
| --- | --- | --- | --- | --- |
| 6.91 | 16.30 | 6605.67 | 4139 | 2125 |

| Circularity index | Elongation ratio | Form factor ratio | Melton ratio | Sinuosity index | Mean Width | Basin relief | |
| --- | --- | --- | --- | --- | --- | --- | --- |
| $Rc = 4\pi A/P^2$ | $Re = \sqrt{(4A/\pi)}/L$ | $Ff = A/L^2$ | | $S = Lcp/L$ | $Wm = A/L$ | $Hr = H - h$ | $Rr = Hr/L$ |
| Miller (1953) | Schumm (1956) | Horton (1932) | Schumm (1977) | | | Hadley and Schumm (1961) | Schumm (1956) |
| 0.33 | 0.45 | 0.16 | | 1.15 | 1046.18 | 2014 | 0.30 |

Table 2