# Peer review of "January 2017. Cordillera Oriental of Argentina"

_Natural Hazards and Earth System Sciences, 2018_

## Referee Comment (RC1) · L. Marchi (Referee) · 23 Jul 2018

Review of the paper NHESS-2018-207

General comment

This paper describes a debris flow that hit a village in northwestern Argentina. This case study could be of some interest to extend the documentation on landslides, debris flows, and alluvial fan flooding in the Quebrada de Humahuaca, which is a region of great interest for the study of these processes. The analysis, unfortunately, is weak and, in my opinion, the present version of this paper is not suitable for NHESS.
My first concern is the general lack of quantification. A more robust and useful work should include the estimation of basic quantitative parameters of the event, such as deposited volume, flow velocity, and peak discharge. Quantitative data provide the basis for choosing and designing debris-flow control measures and permit extending the knowledge of these processes.
Width and thickness of the deposits reported in the paper could permit an estimation, albeit approximate, of debris-flow volume. A large-scale map of debris flow/debris flood deposits would also be useful. The literature, moreover, reports several methods that can be applied for back-calculating flow velocity and discharge from high flow marks, also for non-Newtonian flows. Is there any video recorded by eyewitnesses during the event? Such videos could be very useful to recognize the characteristics of the flow and to reconstruct the velocity.
The rather shallow analysis does not permit to achieve sound results: the comments reported in the conclusions on the studied event, as well as on damage caused by debris flows and debris floods in the Quebrada de Humahuaca and on possible mitigation measures, are very generic.

Specific comments

Los Filtros drains a small catchment: the term 'creek' could be preferred to 'river'.

Page 2, lines 19-21. The two papers cited here deal with flash floods, not debris flows.

Page 3, lines 23-25. Is it relevant for this paper to mention the cement production plant? It is only briefly recalled in the description of damage (page 20, line 3, and figure 2).

Page 4, lines 6-8. The site at which the discharge of the Rio Grande is measured should be reported, with the drainage area of the river basin. Discharge data have little significance unless the drainage area is reported. A better description of the hydrological regime of the Rio Grande would include mean area annual precipitation and mean annual runoff, both in millimeters.

Page 4, lines 21-23. A verbal communication is not acceptable as the only source of rainfall data. Since a recorded rainfall value ('170 mm') is apparently available, the reader would expect to know how and where it was measured. Is there a rain gauges network in the study area? Is there any rain gauge in the Los Filtros catchment? If not, where is located the closest rain gauge?

Page 5, lines 14-15

More could be said about the interaction of the deposits of Los Filtros debris flow/debris flood with the Rio Grande. How was the Rio Grande level when the Los Filtros debris flow occurred (low flow, normal streamflow, flood)? The figure 3g-3h shows little disturbance to the Los Filtros deposits: this could indicate that no flood was occurring in the Rio Grande, but a clear statement is advised.

Page 5, line 16. The section 2.2, in addition to the geological setting, describes also the vegetation cover.

Page 6, line 8. Very generic sentence: it could be removed.

Page 7, lines 5-8.
It is not necessary to present several catchment shape factors: it is clear enough that the Los Filtros catchment has an elongated shape.

Pages 7-8, section 4.1
The alluvial fan slope could provide a useful element for the classification of the catchment-fan system with regard to flow processes (debris flows, debris floods, streamflow with bedload). In addition to the papers by Jackson et al. (1987) and Wilford et al. (2004), the study of Bertrand et al. (2013) could be considered. A valuable feature of the study by Bertrand et al. (2013) is that it takes into account data collected in a number of geographical regions: the thresholds proposed for discriminating debris-flow fans from fluvial fans are thus less depending on local conditions than those of studies developed in one single region (Canadian Rocky Mountains in Jackson et al., 1987 and Wilford et al., 2004).

Pages 8-9, section 4.1
The description of the deposits is used to infer the physical characteristics of the flow (from non-plastic debris flows in the upper part of the channel, to plastic debris flow in the downstream channel reaches). Which are the factors that caused these changes in flow characteristics? A more detailed analysis of sedimentological characteristics of the deposits (including particle size curves) is recommended. This issue would deserve to be discussed also with reference to the literature on debris-flow deposits.

Page 9, lines 14-24.
Debris-flow velocity values are usually reported in m/s.
The values reported are realistic, but their extrapolation from another catchment is questionable. Although the paper by Harrington (1946) includes the name 'Volcan' in the title, it actually refers to the neighboring, and much larger, catchment of the Arroyo del Medio. It is not straightforward to extrapolate a value of debris-flow velocity from a catchment to another, even if they are in the same geographical area. It is also well-known that debris flow velocity may substantially vary in the same channel from event to event, and also during the same event.

Page 11, line 8. 'approximately 76% of the study area is covered by dense vegetation': this statement does not agree with the description of vegetation cover in section 2.2 (page 6, lines 3-4). The lack of dense vegetation is reported on page 13, lines 1-2, and arises from the aerial/satellite images of the catchment (Figs. 1, 6, and 7).

Page 17, line 16. 'Marcato'

Structure of the paper, tables, and figures

Page 4, line 21 – page 5, line 15.
This part of the text belongs to the description of the event, not to the presentation of the study area.

Table 2 is poorly formatted and does not read well.

Fewer photos would be sufficient to document the damage produced by the debris flow/debris flood in the Volcan village, whereas some more photos could be presented to show the morphological and sedimentological characteristics of the deposits. The photos of the deposits could be separated from those of the debris-flow damage to better discriminate the recognition of flow processes from the description of the impacts on urban areas and structures.

References

Bertrand, M., Liébault, F., Piégay, H.: Debris-flow susceptibility of upland catchments. Nat. Hazards 67, 497–511. http://dx.doi.org/10.1007/s11069-013-0575-4, 2013.

Jackson, L.E., Kostashuk, R.A., and MacDonald, G.M.: Identification of debris flow hazard on alluvial fans in the Canadian Rocky Mountains. In Debris flows/Avalanches Process, Recognition, and Mitigation. Geol Soc Am 115-124, 1987.

Wilford, D.J., Sakals, M.E., Innes, J.L., Sidle, N.C., and Bergerund, W.A.: Recognition of debris flow, debris flood and flood hazard through watershed morphometrics. Landslides, 1(1):61-66. doi: 10.1007/s10346-003-0002-0, 2004.

---

## Author Comment (AC1) · 1 Aug 2018

Thank you very much for your corrections.They constitute a major contribution to improving the work and many have already been made. However, the lack of quantification in the study region is due to the lack of rain gauges and other instruments that allow measurements on the other hand, we will close the open discussion of our manuscript due to lack of funds to meet the costs of publication